# Physicochemical, Oxidative Stability and Sensory Properties of Frankfurter-Type Sausage as Influenced by the Addition of Carrot (*Daucus carota*) Paste

**DOI:** 10.3390/foods10123032

**Published:** 2021-12-06

**Authors:** Faisal Eudes Sam, Teng-Zhen Ma, Richard Atinpoore Atuna, Rafia Salifu, Bilal-Ahmad Nubalanaan, Francis Kweku Amagloh, Shun-Yu Han

**Affiliations:** 1Gansu Key Laboratory of Viticulture and Enology, College of Food Science and Engineering, Gansu Agricultural University, Lanzhou 730070, China; sameudes0@gmail.com (F.E.S.); salifurafiat@gmail.com (R.S.); lzhansy@126.com (S.-Y.H.); 2Department of Food Science and Technology, University for Development Studies, Tamale P.O. Box 1882, Ghana; ratuna@uds.edu.gh (R.A.A.); alhassanbilalahmad94@gmail.com (B.-A.N.)

**Keywords:** antioxidant, carrot, frankfurter-type sausage, lipid oxidation, physicochemical properties, sensory characteristics

## Abstract

This study examined the addition of carrot paste (CP) at levels of 3%, 5%, and 10% as a potential antioxidant in frankfurter-type sausages, denoted as F1, F2, and F3. F0, was a control sample with no addition of CP. All formulated samples were stored for 14 days during which their physicochemical, oxidative stability, and sensory properties were evaluated. Results showed that the pH of frankfurter-type sausages was not affected by the addition of CP, however, higher pH values were observed in CP-enriched samples on the first day of production and subsequent storage days. Cooking loss (CL) in frankfurter-type sausages was in the range of 2.20% to 2.87%, with the CP-enriched samples having a lower CL percentage, particularly F3 samples, compared to the control. Protein and fat content were lower in CP-enriched samples, but ash content increased. CP-enriched frankfurter-type sausages recorded significantly higher polyphenol contents compared to the control. Total polyphenol content in CP-enriched samples F1, F2, and F3 was higher throughout storage compared to the control. Lower peroxide values were also recorded in CP-enriched samples F1 (2.5 meq/kg), F2 (2.4 meq/kg), and F3 (2.2 meq/kg) compared to the control (2.9 meq/kg), demonstrating greater 2,2-Diphenyl-1-picrylhydrazyl (DPPH) antioxidant activity than the control samples. Formulations treated with 10% CP gained significantly higher scores for color, texture, and overall acceptability. Principal component analysis showed that higher inclusion levels of CP in formulation improved the sensory quality and oxidative stability. In conclusion, CP could be used to enhance the oxidative stability of frankfurter-type sausage without negatively influencing the sensory quality.

## 1. Introduction

Consumers demand for healthier meat products with better nutritional information have prompted the reformulation of traditional meat products [1,2], including burgers [3,4], meat nuggets [5], and sausages [6,7]. Reformulation of meat products currently involves the use of natural ingredients such as grains [8,9,10], fruits [11,12], vegetables [13,14,15,16], and flower and vegetable oils [4,7,17] that are well-known for their nutritional and medicinal value. The addition of natural ingredients also gives a chance to improve the ‘image’ of some meat products (such as frankfurter-type sausages), which are often regarded as unhealthy by some consumers because of their high fat, cholesterol, sodium chloride, and nitrite content [17,18].

The frankfurter-type sausage is a meat product that attracts consumers from diverse cultures [2,19], owing to its popularity and cost, which is often a driving factor in consumer decisions. Compared to other meat products such as the fresh Italian sausage and fresh bratwurst sausage, some schools of thought suggest that its preparation for consumption takes a relatively short time. In certain circumstances, it simply has to be reheated [2,20]. The frankfurter-type sausage, on the other hand, has a high-fat content of about 20% to 30% in its composition [2], making it susceptible to lipid oxidation [21], resulting in deterioration [22], rancidity [23], discoloration [24], reduced shelf life, and the production of toxic compounds [25]. This has demanded the introduction of antioxidants during the production process to avoid the negative consequences of lipid oxidation.

In the sausage production business, synthetic antioxidants such as butylated hydroxytoluene (BHT), butylated hydroxyanisole (BHA), propyl gallate (PG), and ascorbic acid (AA) have primarily been used to control lipid oxidation. For example, in a range of lipid-based processed products (frankfurter sausage and food oils), AA has been used to prevent lipid oxidation, discoloration, and spoilage in these products [26]. Nonetheless, these synthetic antioxidants have serious health consequences. For instance, according to the European Food Safety Authority Panel on Food Additives and Nutrient Sources added to Food [27], BHT added to food can cause hemorrhage and reduce blood prothrombin index. Botterweck et al. [28] also reported the ability of BHA to cause cancer as well as allergic reactions in sensitive individuals. Furthermore, the frequent intake of AA (1000–2000 mg of ascorbic acid per day) has been reported to increase the potential risk of kidney stones [29,30]. Therefore, great efforts to reformulate healthy meat products with natural antioxidants to replace synthetic compounds are needed.

Plant bioactive compounds such as alkaloids, phenolics, carotenoids, and glucosinolates have been shown to possess antioxidant activity [31,32]. As a result, studies on the use of plants with such bioactive compounds as possible antioxidants in meat products have been conducted [4,13,23,31]. However, in some of these studies, the use of plants with phytochemicals in the formulation of meat products has some drawbacks, such as the replacement of fat or meat, which can result in a less dense matrix microstructure [13] and a lower textural profile [23].

Edible carrot (*Daucus carota*), a root vegetable that ranks among the world’s top ten vegetable crops [33,34], has varying flesh color and different functionality. Edible carrots have white, black, purple, yellow, orange, red, or rainbow roots [35,36]. They contain phytochemicals such as phenolic compounds, carotenoids, polyacetylenes, ascorbic acid [35,37,38], and a distinctive flavor due to the presence of polyacetylenes and terpenoids, making them delicious and healthy [39]. According to Singh et al. [40], colored tropical carrots: black, rainbow, red, orange, and yellow have a total phenolic concentration of 268.08 mg, 41.66 mg, 16.06 mg, 14.52 mg, and 10.82 mg GAE/100 g fresh weight, respectively. Several studies have been conducted to investigate the use of carrot in formulating meat products either as a functional ingredient [16,41,42,43,44], an extender [15,45,46], a dietary fiber [47,48], or a fat replacer [49,50].

Given carrot’s bioactive potential, it could be an excellent replacement for synthetic antioxidants used in meat products, such as frankfurter-type sausage, rich in fat [2] and prone to lipid oxidation [21]. Although few studies have reported on the use of carrot (in the form of powder, paste, or juice) as an antioxidant in the formulation of meat products [13,51,52], there is a paucity of data on the utilization of carrot paste in the development of frankfurter-type sausages. Particularly, the use of carrot paste as a potential antioxidant in the formulation of frankfurter-type sausages without the replacement of fat or meat is rarely reported.

Therefore, this study aimed to incorporate varying levels of carrot paste (CP) in the preparation of frankfurter-type sausages and assess the physicochemical quality, proximate composition, oxidative stability, and sensory quality of these frankfurters.

## 2. Materials and Methods

### 2.1. Materials

About 8.4 kg of lean meat (boneless beef) from the round portion (m. Longissimus Dorsi muscle) and 3.6 kg of pork fat from the brisket and back were used for frankfurter-type sausage formulation. Beef and pork fats acquired from the Meat Processing Unit of the Animal Science Department, University for Development Studies, Ghana, were vacuum-packed in low-density polyethylene pouches, using a tabletop vacuum packer (Telleres Ramon, Vilassar de Dalt, Barcelona, Spain) and kept in a freezer at −18 °C until it was used. The meat was thawed at room temperature (25 °C) for 4 h and minced using a 5-mm-sieve tabletop mincer (Talleres Ramon, Spain) before use. Fresh carrot roots and ingredients (spices) including curing salt, phosphate, red pepper, white pepper, black pepper, and Adobe^®^ were purchased from Tamale Central Market, Northern Region, Ghana. The fresh carrot roots were processed into paste under sanitary conditions using a slightly modified method described by Reddy et al. [15]. In brief, 1 kg of fresh carrot was thoroughly washed under clean running tap water, peeled, cut into 4 mm slices, and blended for 5 min into paste using an electric blender (Robot Coupe^®^, Blixer 3, Clichy, Hauts-de-Seine, France), and stored in a refrigerator at −2 °C until it was used. Later, trials were conducted to produce frankfurters using approximate amounts of minced beef, minced pork, spices, but varying levels of CP (0%, 3%, 5%, 10%, 15% and 20%) to obtain the acceptable incorporation level for further study. Based on preliminary sensory evaluation, the best inclusion levels 3%, 5% and 10% of CP were selected to formulate frankfurters in this study. The 15% addition level was eliminated because it adversely affected some sensory characteristics, particularly flavor, which received low rating as the panelists could perceive raw carrot flavor in its samples.

### 2.2. Design of Experiment

A completely randomized design was used to determine the varying effects of CP and different storage days on frankfurters. The frankfurters were formulated with or without the incorporation of CP. Four different formulations were processed as follows: (i) F0 (control) samples were prepared with 0% CP, (ii) F1 samples were prepared with 3% CP addition, (iii) F2 samples were formulated with 5% CP addition, and (iv) F3 samples were produced with 10% addition of CP. All samples were stored at 4 °C for 14 days.

### 2.3. Frankfurter-Type Sausage Formulation

The frankfurters were formulated according to the method described by Delgado-ospina et al. [6] with slight modification. The frankfurters were prepared using the quantities of the ingredients as shown in Table 1. The given amounts of minced (5 mm) beef and pork fat, CP, curing salt, phosphate, red pepper, white pepper, black pepper, and Adobe^®^ were comminuted in a 3-blade, 30 L-size bowl chopper (Talleres Ramon, Spain) for 5 min at 16 ± 2 °C with the addition of 200 g of ice. A hydraulic stuffer (Talleres Ramon, Spain) was used to stuff the meat paste into natural casings, which were hand-twisted and separated at equal lengths of about 10 cm. The samples were then suspended on smoking shelves, smoked for 45 min, and scalded in hot water (80 °C) to a core temperature of 72 °C in an electric oven (Turbonfan, Blue seal, Birmingham, UK). The samples were then cooled in cold water (15 °C) for 15 min. The cooled samples were hung on shelves for the adhering water to drain and vacuum packaged in low-density polyethylene laminated transparent nylon pouches (Dimensions: 30 × 40 cm, thickness: 51–100 microns, oxygen permeability: 9.3 mL O_2_/m^2^/24 h at 0 °C), and then stored in a refrigerator at a temperature of 4 °C [6,17]. Aliquots of samples were taken for chemical analyses and sensory evaluation tests at an interval of 7 days up to 14 days of storage (4 °C).

### 2.4. pH

The pH was measured according to the method described by Delgado-ospina et al. [6] with slight modification using a pH-meter (WTW pH 330˙I/SET, Germany) equipped with a penetration probe. About 5 g of sample was thoroughly mixed with 50 mL of distilled water using a laboratory mortar and pestle. Before the pH reading, the pH meter was standardized using a buffer solution of pH = 7.02 and pH = 4.00 at 20 °C.

### 2.5. Cooking Loss (CL)

CL was measured using the method described by Grasso et al. [7] where frankfurters were weighed using an electronic scale before and after grilling at 90 °C for 45 min. The cooking loss was calculated as the difference in weight and expressed as a percentage. The same grilling method was employed in preparing samples for sensory evaluation. For each sample, the mean of five measurements was calculated.

### 2.6. Proximate Composition

Proximate analysis to determine the moisture, protein, fat, and ash was carried out according to the AOAC official methods AOAC 925.45, AOAC 981.10, AOAC 991.36, and AOAC 923.03 [53], respectively to establish the nutritive value of the frankfurter-type sausage samples.

### 2.7. Antioxidant Composition

The samples’ total polyphenol (TP) content was determined using the Folin Ciocalteu colorimetric method as described by Blainski et al. [54] with slight modification. Briefly, 200 µL of the sample aqueous extract was mixed with 300 µL 2% Na_2_CO_3_ (Fisher Scientific, Waltham, MA, USA), 50 µL Folin-Ciocalten’s reagent (Sigma-Aldrich Co., St. Louis, MO, USA), and 1 mL of distilled water and left for 1 h at room temperature in total darkness. Gallic acid (Sigma-Aldrich, St. Louis, MO, USA) was used as a standard. The absorbance was measured by Agilent UV–visible spectrophotometer (Agilent 8453, Palo Alto, CA, USA) at 765 nm, and the results were expressed as mg gallic acid equivalents (mg GAE/100 g of the sample).

### 2.8. Measure of Oxidative Stability

The peroxide value (PV) was determined by weighing 1 g of fat from each sample of formulated frankfurter-type sausage into 250 mL Erlenmeyer flasks. Fifteen milliliters (15 mL) of acetic acid–chloroform solution (480 mL Acetic acid and 320 mL Chloroform) was added to each Erlenmeyer flask and swirled. Saturated potassium iodide solution (500 mL) was added to each of the flasks using a 1 mL pipette. Deionized water (20 mL) was added to each flask. They were then titrated against a standard solution of sodium thiosulfate (0.01 N) with 1% starch solution as an indicator. The procedure mentioned above was used without a sausage sample in obtaining the blank. The PV was then calculated and expressed as milliequivalent peroxide per kg of the sample (Equation (1)).
(1)$$\mathrm{PV}\;\left(\mathrm{meq}/\mathrm{kg}\right)=\frac{\left(S\right)\times \;N}{\mathrm{Weight}\;\mathrm{of}\;\mathrm{Sample}\;\left(g\right)}\times 100$$
where: S = Volume of titration, N is the Normality of sodium thiosulfate (0.01 N).

The 2,2-Diphenyl-1-picrylhydrazyl (DPPH) radical scavenging activity was also measured using a method described by Tangkanakul et al. [55], where 100 µL of sample extract was harmonized with 900 µL DPPH (Sigma-Aldrich Co., St. Louis, MO, USA) solution dissolved in ethanol. After 30 min in the dark, the absorbance was measured using spectrophotometry at 517 nm. DPPH radical scavenging activity was then calculated as percent using Equation (2) below.
(2)$$\mathrm{DPPH}\;\left(\%\right)=\frac{\left(1-\mathrm{sample}\;\mathrm{absorbance}\right)}{\mathrm{blank}\;\mathrm{absorbance}}\times 100$$

### 2.9. Sensory Evaluation of Frankfurter-Type Sausages

Sensory evaluation was conducted on samples (F0, F1, F2, and F3) to determine the effect of CP on their sensory characteristics. With a content validated questionnaire, 60 untrained panelists (35 females and 25 males) who regularly consume similar products scored the products based on flavor, color, juiciness, texture, and overall acceptability using a 9-point hedonic scale (where 9—extremely like and 1—extremely dislike) according to the method described by Fernández-López et al. [17]. Before the samples were assessed, samples were grilled with an electrically powered oven (Turbofan, Blue seal, UK) to a core temperature of 90 °C for 45 min, cut into 2 cm thickness, wrapped with coded aluminum foil, and offered to the panelists in a regulated lightening environment. Water was provided to the panelists to be taken as a neutralizer in between products.

### 2.10. Statistical Analysis

Each experiment was conducted with 3 replicates unless otherwise indicated. All the experimental data were analyzed with XLSTAT statistical software package version 2016 (Addinsoft, New York, NY, USA). One-way analysis of variance (ANOVA) was used to determine the effects of formulations and storage period separately, followed by Tukey Pairwise comparison at 5% significance. Furthermore, principal component analysis (PCA) was carried out on the sensory attributes and lipid oxidation/stability of the frankfurters.

## 3. Results and Discussion

### 3.1. Physicochemical Analysis

#### 3.1.1. pH

The pH value is an essential physicochemical parameter used to determine meat and meat products’ quality and shelf life. Generally, the pH of frankfurters was not significantly affected by the addition of CP. However, higher pH values were observed in CP enriched frankfurters on the first day of production and throughout storage although pH decline was observed during storage (Figure 1). The pH of the CP-enriched samples ranged from 5.63 to 5.77, while the control ranged from 5.61 to 5.69.

During storage, a slight decrease in pH was observed for all the samples on days 7 and 14 with CP-enriched frankfurters (mainly, F3 samples) found to have higher pH values compared with the control. Nevertheless, the differences in pH during storage was not significant. The decrease in pH in enriched samples may be due to the acidity of the CP-enriched [15,44,45]. A decrease in pH in chicken nuggets incorporated with mashed sweet potato and ground carrot was also reported [44]. A decrease was also found in turkey sausages as the incorporation level of CP increased from 5 to 15% replacing lean meat [15]. Furthermore, Grasso et al. [7] noticed a reduction in pH of frankfurters formulated with defatted sunflower seed flour. However, in these studies, the decrease in pH was significant. The non-replacement of meat or fat in this study best explains the dissimilarities between this study and others that reported a significant decrease in the pH among samples. Lower pH in meat and meat products can inhibit microbial activities [56] by creating an acidic environment that is unsuitable for microbial growth and reproduction [57]. From this study, higher inclusion levels of CP resulted in a lower pH, therefore, higher inclusions of CP in frankfurter-type sausage formulation may enhance preservation and storage stability.

#### 3.1.2. Cooking Loss (CL)

The CL was used to evaluate the amount of fluids lost after cooking of samples. From the results, a significant gradual increase in CL was observed in the control samples from day 0 to day 14 of storage, whereas in CP-enriched frankfurters, a gradual decrease in CL was found on day 7, which subsequently increased on day 14 of storage. The CL in all frankfurters were in the range of 2.20% to 2.87%, with the CP-enriched frankfurters having lower CL percentage compared to the control throughout storage (Figure 2). However, F3 samples recorded the least cooking loss among the CP-enriched treatments, followed by F2 then F1. The CL trend displayed in Figure 2 indicates that the amount of CP incorporated has a significant influence on the CL of frankfurter-type sausages. From previous studies, the results of CL were in a range of about 3%, which is regarded acceptable for frankfurter-type sausages [58,59]. The present findings support Reddy et al. [15] who found a decrease in CL due to the addition of CP in turkey sausages. Similarly, Shinde [60] reported a reduction in CL of pork patties due to the incorporation of mashed carrot. The varying contents of moisture and fat could most probably account for the differences in CL among the treated samples [12,61]. These results are further supported by Shand [62] and Choi et al. [63,64] who reported that CL may occur through the loss of fat and moisture, associated with the binding capacity between meat protein, fat, and moisture.

In this study, the CL in CP-enriched frankfurters were lower as the inclusion level of CP in frankfurters increased compared with the control, which may also be attributed to the dietary fiber of the CP as carrot contains about 2.36% to 3.77% total crude fiber [46,48,51]. High amounts of dietary fiber have been reported to cause a decrease in CL of meat products compared with their respective control samples [63,65,66]. Therefore, the inclusion of CP in frankfurter-type sausage formulation may improve cooking yield and reduce production cost.

### 3.2. Proximate Composition

The proximate composition (moisture, crude protein, fat, and total ash) of frankfurters is shown in Figure 3. The moisture content of frankfurter sausage is a crucial physicochemical characteristic that can influence sensory attributes such as texture, and affect the shelf life. The moisture content of the treated samples ranged from 63.80% to 67.7% and was significantly affected by the inclusion of CP. The moisture content was significantly lower in the control product (63.80%) as compared with the CP-enriched frankfurters F1 (66.1%), F2 (66.92%), and F3 (67.61%). This finding corroborates earlier studies by Angnihotri and Pal [67], who reported that the moisture content of sausage is about 66.7%. The high moisture content observed in CP-enriched samples was most likely because of the inclusion of fresh CP in the formulation, which added a substantial amount of moisture and fiber to the products as carrot contains about 2.36% to 3.77% total crude fiber [46,48,51]. Higher amounts of dietary fiber in meat emulsion products can increase moisture content due to high water retention of the fiber [68], consequently enhancing the stability of emulsion [69].

Similarly, Bhosale et al. [44] and Kaur et al. [70] found that increasing the level of carrot addition in chicken nuggets caused an increase in moisture content. Zargar et al. [45] also reported a substantial increase in the moisture content of chicken sausages with increasing levels of minced carrot addition, which they ascribed to the higher moisture content of carrot. Furthermore, Reddy et al. [15] and Ahmad et al. [13] reported an increase in moisture content of turkey meat sausages and chicken meat sausages, respectively, with higher inclusion levels of carrot.

Protein plays a crucial role in the water holding capacity of meat due to a strong association of water molecules with meat hemoproteins [71]. A gradual decline in protein content in frankfurter-type sausages was observed as the addition level of CP increased. However, it was insignificant among all the treatments with the control product having a slightly higher protein content than the CP-enriched samples. Although no significant differences were observed among the CP-enriched samples, F1 samples had the highest protein content. Kaur et al. [70] reported that a rise in carrot incorporation caused protein content to decrease. Similar trends were observed in chicken sausages formulated with minced carrot [45]. Reddy et al. [15] reported almost similar findings from a study where they observed that the protein content of CP-enriched (up to 10%) turkey meat decreased insignificantly compared to the control, however, up to 15% level of addition, a significant decrease was observed.

The inclusion of CP in the sausage formulation significantly affected the fat content with control samples having the highest, corresponding to 41% of the total fat content. A slight reduction in fat content was observed as the addition level of CP increased (Figure 3). Among the CP-enriched samples, F3 accounted for the lowest (19.9%), which was significant. However, between F1 and F2, no statistical difference was observed, although the fat content in F1 was higher than F2. Similar findings were reported by Reddy et al. [15] in carrot paste incorporated turkey sausages and by Zargar et al. [45] in carrot incorporated chicken sausages. Moisture is a known diluent of nutrients and as such, tends to reduce the contents of various nutrients when in high amounts. For instance, moisture and fat content are so closely associated with meat products that if the moisture content is high, the fat content is most likely to be lower [72,73]. The decrease in fat content of CP-incorporated frankfurters could be attributed to the addition of fresh CP, which increased moisture content and, as a result, caused a decrease in fat content. The reduction in CP-enriched frankfurters fat content is a positive result as these frankfurters are more likely to undergo lipid oxidation very slowly.

Ash content is a good indicator of total mineral content. The results of this study showed a non-significant (*p* > 0.05) effect on the ash content of the sausages. However, it was observed that increasing inclusion levels of CP resulted in a slight increase in ash content (Figure 3). Similarly, Ahmad et al. [13] found a higher amount of ash in chicken sausage mixed with various vegetables (oyster mushroom, purple cabbage, spinach, capsicum, and carrot), which they attributed to the high fiber content of the vegetables. In contrast, Reddy et al. [15] recorded a decline in ash content in turkey sausages produced with carrot paste. Also, a decrease in ash content in chicken nuggets formulated with carrot was reported by Kaur et al. [70]. The gradual increase in ash content in CP-enriched frankfurters found in this study could be ascribed to individual minerals such as calcium, phosphorus, sodium, potassium, and magnesium that have been reported to be found in carrots [74,75]. Therefore, these minerals may have contributed to the increment in ash content of the frankfurters formulated with CP addition.

### 3.3. Antioxidant Composition

The TP content in the studied frankfurters was determined (Figure 4) and ranged from 51 mg GAE/100 g to 103 mg GAE/100 g of sample. Despite the absence of CP, a value was obtained for the Control samples (formulation without CP addition). It is known that in animal-derived products, so many interfering compounds may provide a reaction to the Folin assay as observed in minced fish [76]. Nonetheless, the most important finding was that on day 0 of production, TP content increased significantly with an increasing level of CP compared with the control. This was anticipated because carrot has been reported in the literature to be a rich source of phenolic compounds [13,37,38,40], which possibly contributed to the higher TP content in the CP-enriched frankfurters as revealed by the results.

Polyphenols from a variety of plants, including carrot, may be able to prevent oxidation in meat products, as in some plants polyphenols ability to delay lipid oxidation in meat products was reported [7,13,21,77,78]. The TP content of the frankfurters decreased over time, with significant variations observed between the CP-enriched frankfurters (F1, F2, and F3) and the control samples (F0), but a non-significant difference was found among the CP-enriched frankfurters, with sample F3 having the highest TP content throughout the storage period. The decrease in TP concentration during storage in both control and CP-enriched frankfurters could be ascribed to the compounds provided by the formulation’s smoke or those naturally present in the frankfurters, both of which are more unstable than the ones added to the frankfurters via the CP. This behavior was also observed in other studies involving various phenolic compounds [21,79]. Based on these findings, CP demonstrated a high potential as an antioxidant source, implying that CP-derived frankfurters would have good oxidation stability.

### 3.4. Measure of Oxidative Stability

Peroxide value (PV) is used as a marker of basic lipid oxidation in meat products to show the rate at which the primary lipid oxidation products are formed. From Figure 5a, the CP-enriched frankfurters had significantly lower PVs than the control sample, possibly due to their high TP content (Figure 4). The TP may have resulted in the scavenging of lipid radicals at the initiation or propagation stages of lipid oxidation reactions, and the formation of low-energy antioxidant radicals in the frankfurters, which cause the oxidation of unsaturated fatty acids [80,81]. Previous studies have found a direct link between TP and antioxidant potential, with TP having the ability to delay lipid oxidation in meat products [7,13,21,40,77,78,82,83]. Among the CP-enriched frankfurters, F3 had the lowest initial PV, followed by F2 and F1. Furthermore, there was a gradual increase in PV as the storage period progressed from day 0 to day 14 of storage. However, primary lipid oxidation was significantly higher in the control compared to the CP-enriched frankfurters. As the storage progressed, the PV of control frankfurters increased significantly almost two-fold from day 0 to day 14, showing the most noticeable lipid oxidation in this sample. Ahmad et al. [13] reported similar results in chicken sausages containing carrot and other vegetables (oyster mushroom, purple cabbage, spinach, and capsicum). This result was expected given that frankfurters (F0; control) without CP addition had the lowest TP content (Figure 4). However, at the end of the storage period, the PV of frankfurters F1 and F3 was the lowest. The rise and fall in PVs could be that the rate at which peroxides and hydroperoxide compounds were formed exceeded the rate at which these compounds decomposed during the first stage of lipid oxidation [84]. According to Dobarganes and Velasco [85], the ongoing formation of hydroperoxides is unstable and can degrade into a variety of volatile and non-volatile compositional products. As a result, hydroperoxides formed as initial oxidation compounds in samples F1 and F3 may have decomposed into secondary oxidation substances. A reduction in the level of primary oxidation compounds is linked with hydroperoxide degradation, which results in secondary lipid peroxidation compounds [86]. Overall, the results showed that incorporating CP provided the best anti-oxidative effect in the CP-enriched frankfurters, improving oxidation stability. Moreover, during the 14 days of storage, all frankfurters produced had PV below 25 meq of active O2/kg fat, which is the acceptability threshold for oily foods [87,88].

The DPPH test was used to evaluate the antioxidant activity of the CP-enriched frankfurters. On the first day of production, the antioxidant activity of frankfurters ranged from 42.3% to 50% and ranged from 36% to 60% after 14 days of storage (Figure 5b). On day 1 of production and during storage, CP-enriched frankfurters (F1, F2, and F3) had a significantly (*p* < 0.05) higher percentage of antioxidant activity than control frankfurters (F0), with sample F3 having the highest DPPH percentage among the CP-enriched samples. This finding is consistent with the TP results (Figure 4).

Several studies have linked phenolic compounds, tannins, and carotenoids to free antiradical activity [13,21,41,78,89,90,91,92,93]. As a result, the higher TP content of the CP-enriched frankfurters best explains the higher antioxidant activity in CP-enriched frankfurters when compared to control frankfurters. Therefore, the findings of this study confirm that the antioxidant activity of CP-enriched frankfurters is influenced by phytochemicals such as polyphenols, carotenoids, polyacetylenes, and ascorbic acid [35,37,38].

### 3.5. Sensory Evaluation of Frankfurters

The sensory quality of the treatments F0, F1, F2, and F3 were evaluated on day 0 of production and days 7 and 14 of refrigerated storage (4 °C). The data (Table 2) in the current study shows that, adding CP to frankfurters significantly (*p* < 0.05) improved the color score compared to the control, with sample F3 (10% addition of CP) ranking the highest. This could be ascribed to the addition of carrot which contains orange color pigments and carotenoids that impart color to the product, which is consistent with the findings of Bhosale et al. [44] and Reddy et al. [15].

During day 14 of refrigeration (4 °C), a slight significant decrease in color scores for all frankfurters was observed. This decline in color scores may be due to changes in meat myoglobin pigment and non-enzymatic browning caused by lipid oxidation compounds reacting with amino acids [48,94]. A reduction in the color scores of carrot-incorporated chicken nuggets as the storage time progressed was also reported by Kaur et al. [70]. Kandeepan et al. [95] also found that color scores in buffalo meat keema decreased as the storage time progressed. Despite the decreasing trend of color scores during storage, the scores of all frankfurters were within the acceptable limits with CP-incorporated frankfurters recording higher mean scores than the control samples.

Aroma and flavor are the most important qualities that affect the sensory properties of comminuted meat products. The addition of CP to frankfurters decreased the flavor ratings significantly as compared to the control on day 0 of storage. Among the CP-enriched frankfurters, the addition of CP up to 5% did not affect flavor strength, but up to 10% addition significantly (*p* < 0.01) reduced flavor ratings, which could be due to the raw carrot flavor [15,44]. The current results are in agreement with those found by Devatkal et al. [96], who discovered a significant decrease in the overall flavor ratings of liver-vegetable buffalo loaves produced with the addition of CP as an extender. Bhosale et al. [44] also found a similar result in chicken nuggets containing up to 15% CP. Reddy et al. [15], on the other hand, found no significant difference in flavor intensity in turkey meat sausages incorporating CP up to 10%, but a decrease was observed when CP was incorporated up to 15%. During subsequent storage days until day 14, flavor ratings for all frankfurters declined significantly except for the control whose decline was insignificant. However, all CP-enriched frankfurters flavor ratings were within the acceptable range (between 7 and 8, that is, “moderately like” and “like very much”, respectively).

As compared to the control, the texture of frankfurters with CP addition improved significantly (*p* < 0.05) on day 0 of storage in the order F3 > F2 > F1 > F0, whereas no significant differences were found between F1, F2, and F3. This may be due to the ability of added fiber (from the CP) to bind water and form gels, which has special characteristics in improving texture. Reddy et al. [15] found similar results for turkey meat sausages containing carrot paste. A detectable slight decrease in texture ratings was found for all the frankfurters during the storage period without significant differences (*p* > 0.05) but CP-enriched frankfurters still recorded higher texture scores than the control samples. Nevertheless, all texture ratings were within a highly acceptable range (8 to 9 where refers to “like very much” and 9 refers to “extremely like”).

The sensory scores for juiciness on day 0 of storage showed that increasing CP incorporation levels substantially increased juiciness. On day 0 of storage, F3 samples differed significantly from F0, F1, and F2 frankfurters, owing to a higher inclusion level of CP (10%) that imparted more moisture compared to F0 (0% CP), F1 (3% CP), and F2 (5% CP) that had lower inclusions of CP with lower moisture content. These results are consistent with those of Kaur et al. [70] for carrot incorporated chicken nuggets and Reddy et al. [15] for CP-enriched turkey meat sausages. Furthermore, Zargar et al. [45] found increasing mean juiciness scores with increasing levels of incorporation of carrot in chicken sausages. Contrary, a non-significant difference was found in chicken nuggets formulated with 10% addition of raw carrot and mashed potato [44] and vegetable-liver loafs produced with the addition of 40% carrot and potato [97]. As the storage days progressed, a significant decrease in juiciness scores for all frankfurters was observed which could be attributed to the gradual loss of moisture from the products. The findings were in agreement with the results of Kaur et al. [70] and Bhat et al. [98] who also reported a decline in the juiciness scores of chicken nuggets and meat balls respectively, during refrigerated storage.

With increasing days of storage, mean scores for overall acceptability showed a significant reducing trend. The 10% CP-enriched samples earned significantly higher overall acceptability scores than the other samples on day 0 of storage. For turkey meat sausages containing up to 10% CP, Reddy et al. [15] found a similar overall acceptability result. The decline in color, flavor, texture, and juiciness scores in all frankfurters during subsequent storage days may be a reflection of the decline in overall acceptability scores. Similar findings were made in chicken seekh kababs [99], fish curls [100], and chicken nuggets [74]. According to the results, CP can be added to the frankfurters’ formulation up to 10% without adversely affecting sensory quality during refrigerated storage.

### 3.6. Principal Component Analysis (PCA)

The differences in the four samples of frankfurters for sensory attributes and lipid oxidation/stability, as well as the similarities between the four samples of frankfurters and each variable, were determined using principal component analysis (PCA), which provides a reduced-dimension plot for data visualization (Figure 6). The first principal component (PC1) accounted for 91.31% of the differences, while the second principal component (PC2) accounted for 8.04%. Texture, color, juiciness, and overall acceptability were the major components contributing to the sensory consistency of CP-enriched frankfurters F2 and F3, while TP and DPPH were the major components contributing to lipid stability. The CP-enriched frankfurters F2 and F3 were clustered together in the same right quadrant of PC1 for those parameters, indicating that higher amounts of CP inclusion in the formulation would improve sensory quality and oxidative stability when compared to frankfurters made without CP or with a lower amount of CP. On the other hand, the control samples were strongly correlated with PV near the negative axis of PC1 and PC2 (lower left quadrant), suggesting their sensitivity to lipid oxidation, which was consistent with PV findings (Figure 5a).

## 4. Limitations

The findings of this study are limited to some physicochemical parameters, oxidative stability indexes, and sensory characteristics which are considered adequate in establishing the nutritional properties and oxidative stability of frankfurter sausage. However, it is possible that other measures of nutritional properties (such as fiber content since carrot is a high-fiber raw material), oxidative stability (including thiobarbituric acid reactive substances (TBARS) to monitor lipid oxidation over time), and microbial stability (microbial analysis) would provide more information in establishing the nutritional quality, oxidative strength, and microbial stability of the formulated frankfurter-type sausage.

## 5. Conclusions and Recommendations

The current study showed the successful utilization of carrot paste (CP) in the formulation of frankfurter-type sausages up to 10% without adverse effects on the physicochemical parameters. CP-enriched frankfurter-type sausages had higher ash content and lower fat content. Moreover, CP-enriched frankfurter-type sausages had lower CL per-centage, particularly formulations with 10% CP addition compared to the control. Furthermore, CP addition resulted in products with more oxidative stability, where CP-enriched sausages recorded significantly higher polyphenol contents than the control. Additionally, the total polyphenol content in CP-enriched samples was higher throughout storage compared to the control. Lower peroxide values were also recorded in CP-enriched samples compared to the control, demonstrating greater 2,2-Diphenyl-1-picrylhydrazyl (DPPH) antioxidant activity than the control frankfurters. As regards sensory quality, no deteriorating effect on the sensory attributes and acceptability of the products formulated with CP addition was observed. Most importantly, formulation with 10% CP addition gained significantly higher scores for color, texture, and overall acceptability. Principal component analysis showed higher inclusion levels of CP in formulation improved the sensory quality and oxidative stability. Therefore, CP could be used to enhance the oxidative stability of frankfurter-type sausage without negatively influencing the sensory quality.

Based on our limitations, we recommend that further research should focus on the nutritional quality, oxidative stability, and microbial stability of the formulated frankfurter-type sausage, taking into account other measures of lipid oxidation, particularly, fiber content, TBARS and microbial analysis. In addition, the storage days should be extended to investigate the shelf life of the product. Furthermore, the use of carrot in other meat products, especially in dry powder form, should be explored.

## Figures and Tables

**Figure 1 foods-10-03032-f001:**
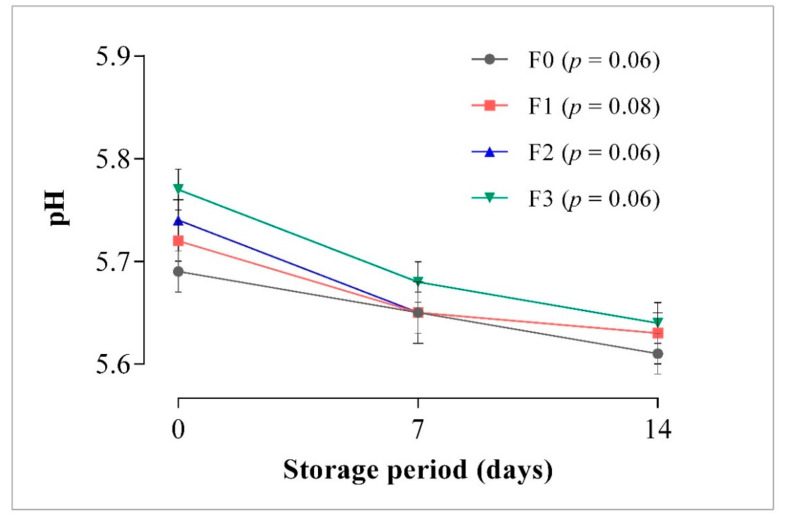
The pH of carrot paste-enriched frankfurters stored at 4 °C, for 14 days. Data are expressed as means (*n* = 3). Error bars represent standard deviation. Treatments with *p*-value less than 0.05 shows significant (*p* < 0.05) difference. F0: control sausage sample without carrot paste, F1: treatment with 3% carrot paste, F2: treatment with 5% carrot paste, and F3: treatment with 10% carrot paste.

**Figure 2 foods-10-03032-f002:**
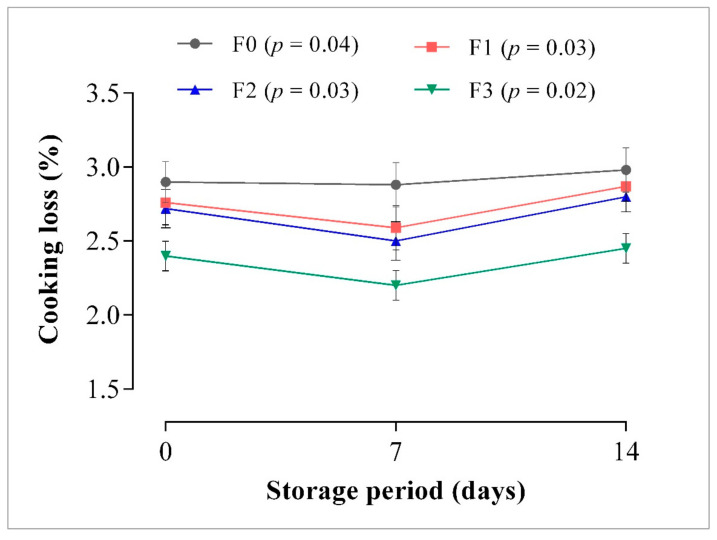
The CL of carrot paste-enriched frankfurters stored at 4 °C, for 14 days. Data are expressed as means (*n* = 3). Error bars represent standard deviation. Treatments with *p*-value less than 0.05 shows significant (*p* < 0.05) difference. F0: control sausage sample without carrot paste, F1: treatment with 3% carrot paste, F2: treatment with 5% carrot paste, and F3: treatment with 10% carrot paste.

**Figure 3 foods-10-03032-f003:**
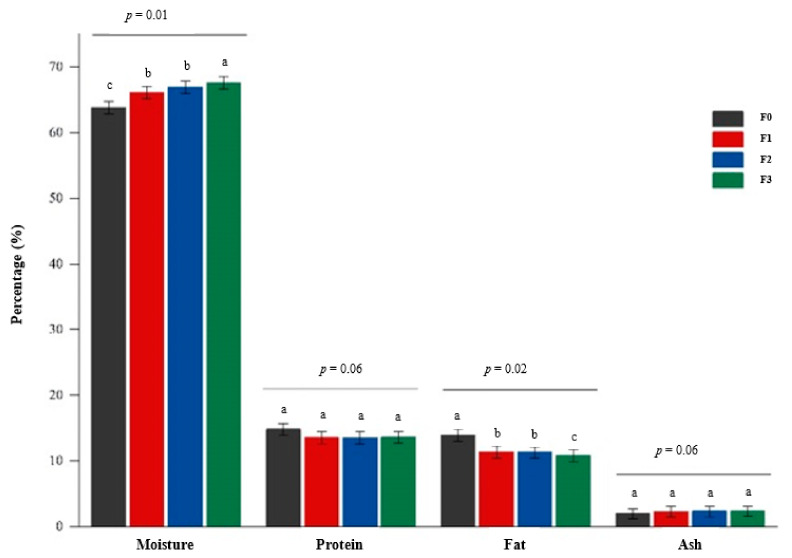
Effects of CP on the proximate composition of frankfurter-type sausages (on dry matter content) on day 0 of storage. Data were expressed as means (*n* = 3). Error bars represent standard deviation. Bars with the same letter are not significantly (*p* > 0.05) different. F0: control sausage sample without carrot paste, F1: treatment with 3% carrot paste, F2: treatment with 5% carrot paste, and F3: treatment with 10% carrot paste.

**Figure 4 foods-10-03032-f004:**
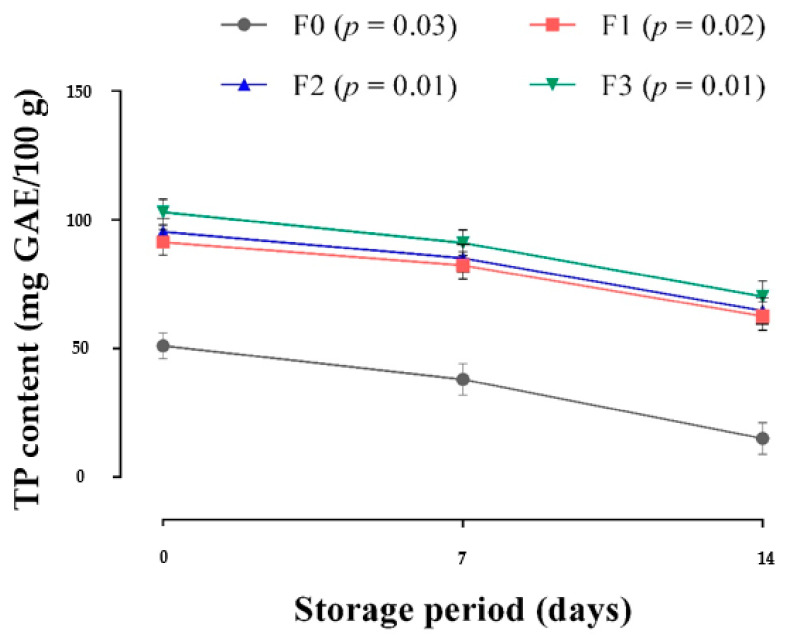
Total phenolic content of frankfurters after 14 days of storage. Data are expressed as means (*n* = 3). Vertical bars represent standard deviation. F0: control sausage sample without carrot paste. F1: treatment with 3% carrot paste. F2: treatment with 5% carrot paste. F3: treatment with 10% carrot paste.

**Figure 5 foods-10-03032-f005:**
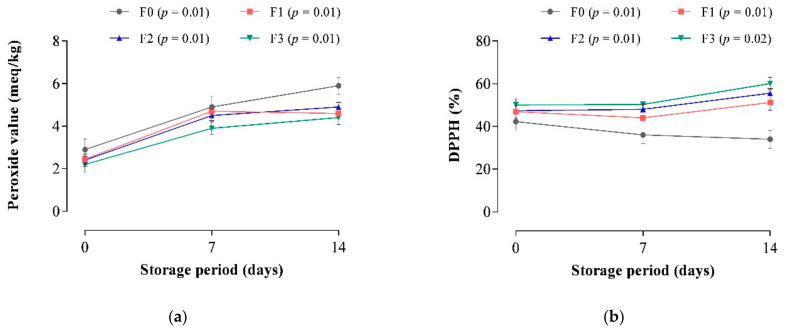
Effect of carrot paste incorporation on; (**a**) peroxide value of frankfurters and (**b**) antioxidant capacity (DPPH) of frankfurters stored for 14 days. Data are expressed as means (*n* = 3). Vertical bars represent standard deviation. F0: control sausage sample without carrot paste. F1: treatment with 3% carrot paste. F2: treatment with 5% carrot paste. F3: treatment with 10% carrot paste.

**Figure 6 foods-10-03032-f006:**
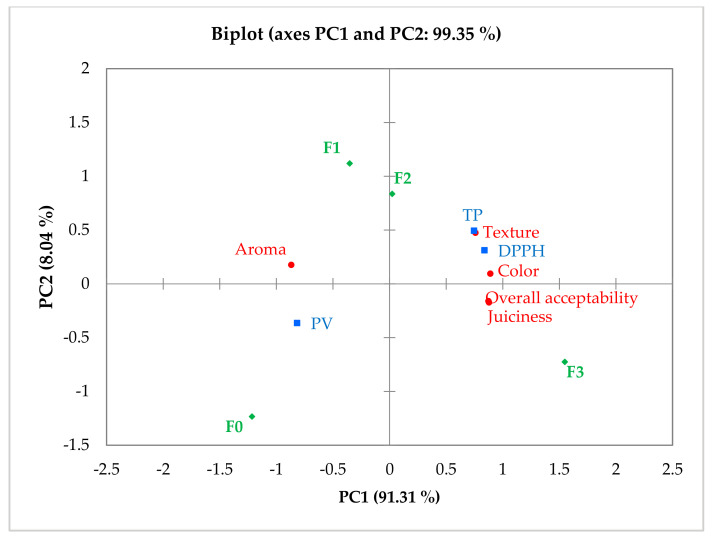
Principal component analysis (PCA) biplot of means of sensory attribute ratings (●) and lipid oxidation/stability (■) of frankfurter-type sausages (◆) enriched with carrot paste. F0: control sausage sample without carrot paste. F1: treatment with 3% carrot paste. F2: treatment with 5% carrot paste. F3: treatment with 10% carrot paste. TP: total polyphenol. PV: peroxide value. DPPH: 2,2-Diphenyl-1-picrylhydrazyl.

**Table 1 foods-10-03032-t001:** Raw materials and their amounts used in the formulation of frankfurter-type sausage.

Ingredients	Samples			
	F0	F1	F2	F3
Lean meat (Kg)	2.1	2.1	2.1	2.1
Pork fat (Kg)	0.9	0.9	0.9	0.9
Ice (g)	200	200	200	200
Curing salt (g)	45	45	45	45
Polyphosphate (g)	15	15	15	15
Red pepper (g)	1.5	1.5	1.5	1.5
White pepper (g)	3	3	3	3
Black pepper (g)	3	3	3	3
Adobe^®^ (g)	6	6	6	6
Carrot paste (g)	0	90 (3%)	150 (5%)	300 (10%)

**Table 2 foods-10-03032-t002:** Sensory qualities of frankfurters during refrigerated storage at 4 °C (mean).

Attribute	Samples	Storage Period (Days)		
		0	7	14
Color	F0	8.19^cA^	8.13^cB^	8.10^cB^
	F1	8.33^bA^	8.23^bB^	8.10b^cC^
	F2	8.38^bA^	8.25^bB^	8.15^bC^
	F3	8.55^aA^	8.46^aB^	8.36^aC^
Flavor	F0	8.49^aA^	8.42^aA^	8.41^aA^
	F1	8.41^abA^	8.33^abB^	7.82^bC^
	F2	8.40^abA^	8.31^abB^	7.83^bC^
	F3	8.10^cA^	7.77^cB^	7.53^cB^
Texture	F0	8.02^bA^	8.01^bA^	8.00^bA^
	F1	8.31^aA^	8.29^aA^	8.27^aA^
	F2	8.33^aA^	8.32^aA^	8.30^aA^
	F3	8.40^aA^	8.38^aA^	8.39^aA^
Juiciness	F0	8.29^bA^	7.00^aB^	6.12^aC^
	F1	8.31^bA^	7.11^aB^	6.11^aC^
	F2	8.33^bA^	7.34^aB^	6.23^aC^
	F3	8.42^aA^	7.37^aB^	6.26^aC^
Overall acceptability	F0	8.24^bA^	7.80^bB^	7.70^bB^
	F1	8.25^bA^	7.91^bB^	7.80^bB^
	F2	8.28^bA^	7.94^bB^	7.82^bB^
	F3	8.34^aA^	7.99^aB^	7.91^aB^

F0: control sausage sample without carrot paste, F1: treatment with 3% carrot paste, F2: treatment with 5% carrot paste, F3: treatment with 10% carrot paste. Means with different superscripts in a row orientation (uppercase alphabet) indicate significant (*p* < 0.05) differences between different storage days at the same CP addition level. Means with different superscripts in a column orientation (lowercase alphabet) indicate significant (*p* < 0.05) differences between different CP addition levels at the same storage day.

## Data Availability

The data presented in this study are available on request from the corresponding author.

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
