# Peer review of "Physicochemical, Oxidative Stability and Sensory Properties of Frankfurter-Type Sausage as Influenced by the Addition of Carrot (Daucus carota) Paste"

_foods, 2021, doi:10.3390/foods10123032_

Round 1

Reviewer 1 Report

In the present manuscript authors describe the investigation of the “Physicochemical, oxidative stability and sensory properties of frankfurter sausage as influenced by the addition of carrot (Daucus carota) paste”.

The introduction is clear and shows the literature data about the man subject. Methodology is adequate.

Results seem fine and appear correct.

References were correctly selected and compatible with the results and the data obtained.

The manuscript can be further processed after minor revision, as shown below:

Abstract:

 Cooking loss in frankfurters was in the range of 2.20% to 2.87%, with the CP-enriched frankfurters having lower CL percentage, particularly F3 samples, compared to the control.

CL acronym has no definition in the abstract, only in the text. The Abstract has to be understood without the need to read the text. Put in front of the words Cooking loss, between parentheses (CL).

Introduction:

Consumer demand for healthier meat products with better nutritional information, as well as the recent COVID-19 outbreak, have prompted the reformulation of traditional meat products, including burgers, meat nuggets, and sausages.

I consider that the theme of the work and even its content does not have a direct relationship with COVID-19, I suggest removing this comment.

Reviewer 2 Report

In this work, the authors investigate the use of carrot paste for the formulation of frankfurter sausages. The article is well written, methods and results are well presented. From that point of view it has worth but some parts need to be improved. 

The other comments are reported in the attached file.

Reviewer 3 Report

The article presents the results of research on the addition of carrot paste at the level of 3%, 5% and 10% to the production of frakneurfer sausages for their sensory quality and oxidative stability.

The article is interesting and presents an interesting solution, although discussed in the literature of the subject, of adding plant additives to meats. In my opinion, the article is technically very reliable. The research is presented in a very simple way, with each method used by the authors a comparison is made with the results available in the literature.

My reservations are only raised by the conclusions which are very laconic and do not show the authors' work. In my opinion, from so many studies carried out, conclusions could be extended, e.g. by collected conclusions from individual stages and presenting them to the reader as concise conclusions.

Reviewer 4 Report

Manuscript #1454439 studies the influence of the addition of carrot paste in properties of frankfurter sausage.

Quite interesting study but I think that the study should be enriched with additional analyses.

  1. The frankfurter sausage has specific recipe or/and slight moderations. The authors have added carrot paste so this product cannot be named “frankfurter”. Maybe frankfurter type sausage?
  2. The sampling for analysis was 1 week and 2 weeks of storage. How did these days have been decided? Only 2 times of analysis can be promote safe data? What about the shelf life of the products?
  3. The most important: Here we have food and sensory analysis has been conducted. How did the authors are sure that the product was safe for consumption? This is very dangerous and microbiological analysis should be done.
  4. Furthermore, when the author tried to justify the results, they must keep in mind that the characteristics of new products will be quite different when the CP is added in meat. i.e. the content fat in new products was lower comparing to control samples so the peroxide value, as it was measured, maybe is not right as in lower fat contents leads in lower PV values. (See following comment).

Some specifics comments:

Lines 78 and 79: “polyacetylenes”?? What are these compounds? Are inherent compounds? Please check out the presence of them in foods. Did the authors mean something else?

Line 137: “… nylon polyethylene bags..” Which kind of nylon? Is this material multilayer, laminated or coextruded? Which was the thickness of each material and the overall thickness of the material?

Line 161: Correct the superscripts in Na2CO3.

Line 168: oxidative stability: This method is a method analysis for oils and fats. In foods we usually use another methods analysis as foods are complicated matrices. These results (in current study) are representative? I do not think so as the sample does not consist only by fat.

Reviewer 5 Report

In order to study oxidation status of meat and meat products, it is important evaluated pro- and anti-oxidant factors. In this study several parameters of oxidation stability were determined, but thiobarbituric acid reactive substances (TBARS) and free fatty acids were not determined which can be more explain the oxidation status of evaluated samples.

Why statistical analysis for indication the significant (p < 0.05) differences between different storage days at same CP addition level not performed for all evaluated parameters like that conducted for sensory qualities

In abstract: mention the method and period and storage of samples

Line 114: minced beef

Lines 155-156: AOAC official method for ash not mentioned

Line 220: the decrease in pH

In figure 3: for protein, although the p=0.04 (p > 0.05), no significant differences were observed between F0a, F1a,F3a,F4a

Line 336: It is known that in…

Line 336: matrixes?

Lines 338, 399, 412: on day 1 of production, but in the resulted (Figure 4, 5 and table 2) there is day 0 of storage.

Line 450-451: significantly (p < 0.05) differences only between F0 and CP addition samples. No significantly differences between F3 > F2 > F1 >.

Line 483: of frankfurters
